# Exogenous Transforming Growth Factor-β in Brain-Induced Symptoms of Central Fatigue and Suppressed Dopamine Production in Mice

**DOI:** 10.3390/ijms22052580

**Published:** 2021-03-04

**Authors:** Won Kil Lee, Yeongyeong Kim, Heejin Jang, Joo Hye Sim, Hye Jin Choi, Younmin Shin, Jeong June Choi

**Affiliations:** Laboratory of Molecular Medicine, College of Korean Medicine, Daejeon University, Daejeon 34520, Korea; lwk2737@dju.ac.kr (W.K.L.); cabby288@naver.com (Y.K.); gmlwls0766@naver.com (H.J.); simjoohye@naver.com (J.H.S.); haejinchoi@hanmail.net (H.J.C.); min050404@naver.com (Y.S.)

**Keywords:** chronic fatigue syndrome, myalgic encephalomyelitis, systemic exertion intolerance disease, transforming growth factor-β, dopamine

## Abstract

Myalgic encephalomyelitis (ME)/chronic fatigue syndrome (CFS) is one of the most refractory diseases in humans and is characterized by severe central fatigue accompanied with various symptoms that affect daily life, such as impaired memory, depression, and somatic pain. However, the etiology and pathophysiological mechanisms of CFS remain unknown. To investigate the pathophysiological role of transforming growth factor (TGF)-β1, we injected a cytokine into the lateral ventricle of a C57BL/6 mouse. The intracranial injection of TGF-β1 increased the immobility duration in a forced swimming test (FST) and time spent at the closed arm in elevated plus maze (EPM) analysis. The mice injected with TGF-β1 into their brain showed increased sensitivity to pain in a von Frey test, and had a decreased retention time on rotarod and latency time in a bright box in a passive avoidance test. In addition, the serum levels of muscle fatigue biomarkers, lactate dehydrogenase (LDH) and creatine kinase (CK), were significantly increased after administration of TGF-β1. Intracranial injection of TGF-β1 significantly reduced the production of tyrosine hydroxylase (TH) in the ventral tegmental area, accompanied by a decreased level of dopamine in the striatum. The suppression of TH expression by TGF-β1 was confirmed in the human neuroblastoma cell line, SH-SY5Y. These results, which show that TGF-β1 induced fatigue-like behaviors by suppressing dopamine production, suggest that TGF-β1 plays a critical role in the development of central fatigue and is, therefore, a potential therapeutic target of the disease.

## 1. Introduction

Myalgic encephalomyelitis (ME)/chronic fatigue syndrome (CFS) is a chronic fatigue disease that is associated with a wide spectrum of symptoms such as chronic fatigue, sleep disorders, impaired memory and concentration, new-type headaches, muscle pain, sore throat, and arthralgias [1]. The worldwide prevalence of CFS is estimated at 0.1–1% [1,2]. According to data published in the United States, CFS occurs in about 0.23 to 0.42% of the population, and more than 75% of the patients were female, with the average age of patients being between 29 and 35 years old [3]. A clear cause of CFS has still not been identified despite long-term research. However, it is thought that lifestyle, disease and accident trauma are the most likely causes. Importantly, there is no proven treatment for CFS, and common treatment strategies are primarily used to limit symptoms rather than treat the underlying cause [4]. Due to the lack of medical knowledge and appropriate treatment, patients with chronic fatigue syndrome suffer with metal stress as well as physical symptoms [5]. Therefore, investigation of the pathophysiology of CFS is urgently required.

Although the cause and pathogenesis of CFS are not exactly understood, the endocrine system, immune system, nervous system, metabolic system and gut microbe system are suggested to be involved in CFS. Recently, it was reported that abnormal immune responses have been increasingly suggested as a causal factor of CFS [6]. An increased amount of pro-inflammatory cytokines was detected in the blood of patients with CFS [7]. The T helper 2 immune response involving IL-4 was reported to be greatly increased in lymphocytes from patients with CFS [8]. Neuro-inflammation activates oxidative and nitrosative stress pathways, resulting in mitochondrial dysfunctions [9]. A positron emission tomography (PET) analysis with (11)C-(R)-(2-chlorophenyl)-N-methyl-N-(1-methylpropyl)-3-isoquinoline-carboxamide showed that neuroinflammation was prominent in the brain area of the cingulate cortex, hippocampus, amygdala, thalamus, midbrain, and pons in patients with CFS [10].

The increased level of transforming growth factor (TGF)-β in patients with CFS suggests that this cytokine has a pathophysiological role in the onset of the disease. A case-control study with 24 patients with CFS and 21 healthy control participants showed that the patients with CFS had a significantly higher level of TGF-β [11]. Another cross-sectional study with 192 ME/CFS patients and 392 healthy controls also showed that a total of 17 cytokines were elevated in the patients [12]. The study showed that only TGF-β was completely elevated in patients with CFS among the elevated cytokines. Bennett et al. reported that the level of circulating bioactive TGF-β was significantly increased in patients with CFS [13]. A meta-analysis study demonstrated that TGF-β, IL-2 and IL-4 were markedly elevated in patients with CFS when compared with those in the healthy control group [14]. These findings of the strong correlation between TGF-β and CFS highlight the possible pathogenic roles of the cytokine in CFS.

TGF-β is a superfamily of cytokines that possesses various physiological functions. The TGF-β signaling pathway plays an important role in regulating vital cellular activities such as cell growth, differentiation, apoptosis, motility, invasion, extracellular matrix production, angiogenesis, and immune response. All three isoforms of TGF-β, TGF-β1, TGF-β2, and TGF-β3 are synthesized in cells as precursor molecules that contain a propeptide region in addition to the TGF-β homodimer. TGF-β1 is generally known to inhibit cancer [15]. However, it is known that overexpression of TGF-β1 promotes tumor growth or metastasis [16,17]. Mutations in the TGF-β1 signaling system receptors or Smad genes are known to be involved in tumor growth inhibition or tumors. TGF promotes wound healing and scar formation. Injury activates macrophages to produce active TGF-β, which stimulates the production of vascular endothelial growth factor (VEGF) to induce angiogenesis [18]. TGF-β activates fibroblast to produce extracellular components such as fibronectin and collagen, which are the basic molecules involved in scar formation [19].

Despite the above-mentioned beneficial functions of TGF-β, the studies on the increased level of the cytokine in patients with CFS proposed that TGF-β plays a role in the pathogenesis of CFS. In this study, we performed an intracerebroventricular (i.c.v.) injection of TGF-β into the brains of mice and investigated the effect of TGF-β on fatigue-like behaviors and the production of neurotransmitters in the brain. Our results suggest that TGF-β1 modulates the behaviors and symptoms related with CFS and that this animal model is applicable in a study of the central fatigue.

## 2. Results

### 2.1. Effects of TGF-β1 on Depressive and Anxious Behaviors

In order to investigate the effects of TGF-β1 on behaviors of depression and anxiety, the mice were intracranially injected with TGF-β1. A forced swimming test showed that TGF-β1 injection significantly reduced the duration of immobilization (*F*(1,12) = 22, Figure 1A). TGF-β1 increased the time that the mice remained in the dark compartment compared with the sham group in an elevated plus maze test (*F*(1,7) = 2.496, Figure 1B). The locomotor activity of the mice that received TGF-β1 injection was decreased compared with that of sham group; however, there was no statistical significance (*F*(1,11) = 1.720, Figure 1C). These results indicate that intracranial injection of TGF-β1 induced depressive and anxious behaviors in mice.

### 2.2. Effects of TGF-β1 on Fatigue-Like Behaviors

Next, we performed the rotarod test to investigate the effect of TGF-β1 on motor activity, which is related to fatigue in CFS. Those mice that received an injection of TGF-β1 into the brain showed decreased latency time on rotarod (*F*(1,8) = 2.137, Figure 2A). This result indicates that TGF-β1 induced fatigue-like behaviors.

### 2.3. Effects of TGF-β1 on Learning Memory

To examine the effects of TGF-β1 on learning memory, a passive avoidance test was performed. When mice were injected with TGF-β1 into the brain, the latency time of escaping from the light box was greatly reduced compared with the sham group (*F*(1,12) = 2.2784, Figure 2B). This result indicates that TGF-β1 reduced the learning memory ability of the mice.

### 2.4. Effects of TGF-β1 on Allodynia

To examine the effects of TGF-β1 on sensitivity to pain, which is one of main symptoms in patients with CFS, a von Frey test was performed. It was observed that the pain score was increased in mice injected with TGF-β1 into the brain compared with the sham group (*F*(1,14) = 2.910, Figure 2C). This result indicates that TGF-β1 elevated sensitization to pain.

### 2.5. Effects of TGF-β1 on Enzyme Markers of Fatigue in Serum

Next, we measured the serum levels of lactate dehydrogenase (LDH) and creatine kinase (CK), which are commonly elevated in chronic fatigue syndrome [20]. The injection of TGF-β1 significantly increased the level of LDH (1850 ± 446 U/L) compared with the sham group (569 ± 150 U/L) (Figure 3A). The serum level of CK was significantly increased (4580 ± 1440 U/L) compared with the sham group (1090 ± 338 U/L) (Figure 3B). Taken together, TGF-β1 in the brain appears to induce fatigue-related biochemical markers.

### 2.6. Effects of TGF-β1 on Dopaminergic System

Dopamine plays a role in transmitting neurons from cranial nerve cells, and the lack of production or secretion of this neurotransmitter is known to cause various neurologic disorders [21]. Here, we investigated the level of tyrosine hydroxylase (TH)—an enzyme that transforms thyroxin to dopamine—in the ventral tegmental area. Immunohistochemistry analysis showed that injection of TGF-β1 significantly reduced the production of the TH compared with the sham group (Figure 4A). Interestingly, TGF-β1 changed the distribution of TH in neuronal cells. The expression of TH upon TGF-β1 injection was concentrated and limited to within the cytoplasm. Immunohistofluorescence analysis with dopamine antibody showed that the level of dopamine in the striatum was decreased upon TGF-β1 injection (Figure 4B). The results indicate that TGF-β1 may induce fatigue-like behaviors in mice by regulating the production and secretion of dopamine.

### 2.7. Effects of TGF-β1 on the Expression of TH in SH-SY5Y Cells

To confirm that TGF-β1 suppresses the expression of TH, we treated SH-SY5Y cells—human neuroblastoma cells possessing features of dopaminergic neurons—with TGF-β1 and measured the level of TH using Western blot analysis and real-time PCR. The administration of TGF-β1 inhibited the expression of TH in both protein and mRNA (Figure 5).

## 3. Discussion

TGF-β plays various roles in the central nervous system (CNS). Brain injury and stroke activate TGF-β in the brain [22]. The function of TGF-β1 is to facilitate the repair of damaged brain and promote scar formation [23]. Angiogenesis is also induced by TGF-β activation for brain repair [24]. However, the production of TGF-β in the CNS can also provide harmful effects. TGF-β from astrocytes were reported to facilitate the production of amyloid-β in the brain, potentially inducing Alzheimer’s disease [25]. The proliferation of neural stem cells in adults is suppressed by TGF-β [26]. Excessive exogenous injection of TGF-β hindered the regeneration of spinal tissue in rats with a spinal injury [27]. Severe exercise induces the accumulation of TGF-β in cerebrospinal fluid (CSF), which mediates central fatigue [28,29]. Recently, Lee et al. reported that a promising CFS animal model, adrenalectomy mouse, showed increased levels of TGF-β in serum and brain areas, including the prefrontal cortex, hippocampus, hypothalamus, and raphe nuclei [30]. Intraventricular injection of TGF-β1 impaired spatial learning memory accompanied with subarachnoid fibrosis and reduced the sodium pump activity in rats [31]. These findings support our hypothesis that TGF-β is a critical player in the pathogenesis of CFS.

We demonstrated that TGF-β alters the production of dopamine, which mediates depression and anxiety disorders. Dopamine plays a number of roles in several brain functions, including important roles in behavior and cognition, sleep, mood, attention, memory and learning [32]. Insufficient production of dopamine is known to result in the loss of motor and muscle capacity and lead to memory, attention and cognitive impairment. This decreased production of dopamine has been linked to major depression in some patients suffering from severe fatigue [32]. Patients with Parkinson’s disease, which is a neurodegenerative disease caused by abnormal dopaminergic signaling in the brain, complain of physical fatigue as a result of the decreased dopamine level [33]. Decreased dopamine production or secretion can play a role in inducing the symptoms of fatigue. Other studies support the involvement of dopamine in symptoms of CFS. A study in patients with chronic fatigue syndrome reported that the level of tyrosine, a precursor to dopamine, was low compared with control [34]. The suppressed activation of the basal ganglia was reported in patients with CFS [35]. Impaired functions of the right caudate and right globus pallidus, a brain region with a high concentration of dopaminergic neurons, were observed. The amount of evidence that dopaminergic pathways are involved with in the pathogenesis of CFS is growing. Based on our results, high and/or chronic TGF-β levels in the brain suppress dopamine production, leading to symptoms of fatigue.

The etiology of increased levels of TGF-β in patients with CFS remains unclear. One possible hypothesis is that the production of TGF-β might be induced by pro-inflammatory cytokines. TGF-β is secreted by macrophages upon the stimulation of pro-inflammatory cytokines. In physiological conditions, the TGF-β secreted by pro-inflammatory reactions balances these inflammatory reactions by suppressing the production of TNF-α, IL-1, and IL-6 [36]. In other words, the high levels of pro-inflammatory cytokines in patients with CFS might induce the production of TGF-β to compensate. The chronic activation of inflammation in CFS may sustain the high level of TGF-β in serum or the brain. However, this hypothesis remains to be confirmed.

Our study investigated the pathogenic role of TGF-β1 in CFS using an animal model. Although it has been well reported that the presence of TGF-β in the blood is associated with CFS, the level of TGF-β in the CNS of patients with CFS remains unclear. However, the studies that report the induction of CFS-like symptoms as a result of TGF-β in the brain imply the pathophysiological role of TGF-β in CFS. This study demonstrated that i.c.v. injection of TGF-β in the brains of mice mimicked clinical features of CFS, such as depression, anxiety, hypersensitivity to pain, and impaired learning memory accompanied with decreased dopamine production. However, the effects of TGF-β on characteristic symptoms of CFS, post-exertional malaise and unrefreshing sleep, remain to be elucidated. Therefore, this study suggests that a potential molecular mechanism that induces central fatigue with an exogenous injection of TGF-β can be a useful animal model for a pharmaceutical investigation of CFS.

## 4. Materials and Methods

### 4.1. Animals

Male C57BL/6 mice (6 weeks old, 19–22 g) were purchased from DBL, Ltd. (Eumsung, Korea). The animals were maintained under standard conditions (12:12 h light–dark cycle, 23 ± 1 °C), with free access to water and food. The mice were subjected to one week of acclimatization before the experiments commenced. All the experiments involving animals were carried out in accordance with the National Institute of Health Guide for the Care and Use of Laboratory Animals (NIH publication no. 85–23, revised 1985), and the Institutional Animal Care and Use Committee of the Daejeon University approved the experimental protocol (DJUARB2020-033, 7 December 2020).

### 4.2. Intracerebroventricular Injection of TGF-β1

The mice were anesthetized by respiratory inhalation of isoflurane solution. The hairs on the heads of the mice were shaved. After fixing the mice on the stereotaxic frame (NEUROSTAR, Tubingen, Germany), a hole was made using a drill (1 mm diameter). TGF-β1 (100 ng in phosphate-buffered saline (PBS) with 0.1% bovine serum albumin (BSA), Perotech, Markham, ON, Canada) was injected with a Hamilton syringe needle (26-gauge) into the position of a lateral ventricle (ML: 0.9, AP: −0.1, DV: 3.0) at a rate of 1 μL/min. The total injected volume per mouse was 5 μL. After the injection, the scalp was sutured using a sterilized suture and a disinfectant was applied. The sham group was treated with an equivalent volume of vehicle (0.1% BSA in PBS) via the same methods used for the TGF-β1 group.

### 4.3. Open Field Test

The open field test (OFT) was conducted 15 days after TGF-β1 injection, as previously described, with minor modification [37]. The plastic enclosure box for the open field apparatus was contained within the black square side (40 cm × 40 cm × 30 cm), and the center of the field was distinguishable in the recording software. Before beginning the trial, mice were acclimated in the testing room for 30 min. Behavioral testing was conducted twice for 5 min (interval 1 h) at 7–8 lux illumination. The movements of animals were recorded using a video camera connected to the corresponding software (Smart Junior, Panlab SL, Barcelona, Spain).

### 4.4. Rotarod Test

The motor activity was evaluated using a rotarod machine (ENV-574M, Med Associates Inc., St. Albans, VT, USA) according to the manufacturer’s instructions at 16 days after TGF-β1 injection. Before the training test, the mice were habituated to stay on the stationary drum (4 rpm, 1.12 m/min on the surface) for 3 min. The mice were placed back on the drum immediately if they fell up to five times during habituation. After habituation for 3 days, the motor activity of the mice was evaluated (three trials, interval 20 min) on the drum, which was accelerated from 4 to 40 rpm. The latency to fall was recorded.

### 4.5. Passive Avoidance Test

Fear-conditioning learning and memory were evaluated using a step-through passive avoidance apparatus (JD-SI-10, JEUNGDO Bio & Plant Co., Ltd., Seoul, Korea) with a two-compartment box (one light and one dark compartment connected by a sliding door) at 16 days after TGF-β1 injection. In the acquisition trial, the mice were subjected to an electrical shock (0.4 mA) for 5 s when they entered the dark compartment. After 24 h, each mouse was placed back into the light compartment for retention trials. The latency time to enter the dark compartment was recorded for 5 min.

### 4.6. Von Frey Test

The sensitivity of the hind paw was measured 15 days after TGF-β1 injection using an electronic von Frey aesthesiometer (IITC Inc. Life Science, Woodland Hills, CA, USA). The mice were placed in a box (10 × 10 × 10 cm^3^) with a wire mesh on the bottom and stabilized for 1 h. The number of lifting or scratching behaviors on hind paws was counted during 10 pokes with a hair (1.3 g von Frey hair).

### 4.7. Sacrifice

The mice were sacrificed under isoflurane anesthesia. The serum was collected by centrifugation at 3000× *g* for 15 min. For the immunohistological analysis, the mice were subjected to transcardial perfusion with heparin (10 units/mL) and 4% paraformaldehyde (PFA) solution, and the isolated brains were fixed with 4% PFA.

### 4.8. Blood Biochemical Examination

Lactate dehydrogenase (LDH) and creatinine kinase (CK) were measured using a blood biochemical analyzer (BS 220, Mindray, Shenzhen, China).

### 4.9. Immunofluorescence Staining

The brain sections were prepared by cutting them to a thickness of 35 μm using a cryostat (CM3050S, Leica Microsystems, Nussloch, Germany). To proceed with immunofluorescence staining, PBS containing 2% Triton X-100 was reacted for 1 h at room temperature. Then, PBS containing 2% normal goat serum was reacted for 1 h at room temperature to suppress a non-specific reaction. Dopamine antibody was diluted 1:400 in PBS containing 2% normal serum and reacted at 4 °C for 18 h. After washing three times with PBS, the secondary antibodies—goat anti-mouse IgG conjugated with TRITC (Invitrogen, Carlsbad, CA, USA)—were diluted to 1:200 in PBS containing 2% normal serum and reacted at room temperature for 2 h. The nuclei were stained with Hoechst (Invitrogen), washed with PBS, and then covered with a cover glass. The tissue slide was observed at 10× using a Nikon ECLIPSE E600 (Nikon, Japan) fluorescence microscope.

### 4.10. Immunohistochemistry Staining

Brain sections were prepared as described in Section 4.7 To block endogenous peroxidase activity, the free-floating sections were immersed in 1% H_2_O_2_. The sections were treated with blocking buffer (5% normal chicken serum and 0.3% Triton X-100 in cold PBS) and incubated with tyrosine hydroxylase (1:200, #SC-25269, Santa Cruz Biotechnology, Santa Cruz, CA, USA) primary antibodies overnight at 4 °C. After washing with ice-cold PBS, sections were incubated with a goat anti-mouse IgG HRP (1:200, Vector Laboratories, BA-1000) secondary antibody for 2 h at 4 °C. The sections were exposed to an avidin–biotin peroxidase complex (PK-6200, VECTASTAIN Elite ABC kit, Vector Laboratories, Burlingame, CA, USA) for 1 h. Then, the peroxidase activity was color-developed with stable 3′-diaminobenzidine. The tissue slide was observed at 40× using a Nikon ECLIPSE E600 (Nikon, Japan) microscope.

### 4.11. Cell Culture

The human neuroblastoma cell line, SH-SY5Y cells (American Type Culture Collection, Manassas, VA, USA) were maintained in DMEM supplemented with 10% heat-inactivated FBS and 1% (*v*/*v*) penicillin/streptomycin at 37 °C in an atmosphere of 5% CO_2_. The medium was replaced every 2–3 days.

### 4.12. Western Blotting Analysis

The SH-SY5Y cells were seeded into a six-well culture plate at a density of 2 × 10^5^ cells/well for 24 h and then treated with TGF-β (1–20 ng/mL) for 24 h. They were then rinsed with ice-cold PBS and lysed in the RIPA buffer. Equal amounts of each protein sample were resolved on 8–18% sodium dodecyl sulfate-polyacrylamide gels; the resultant bands were transferred onto nitrocellulose membranes (Hybond ECL; Amersham Pharmacia Biotech, Piscataway, NJ, USA). The membranes were blocked in 5% skim milk solution for 1 h. Next, they were incubated with antibodies against TH (1:1000; Santa Cruz Biotechnology Inc., Dallas, TX, USA), and TGF-β receptor (1:1000; Abcam, Cambridge, UK) overnight at 4 °C, and then with horseradish peroxidase-labeled IgG antibodies (1:2000; Santa Cruz Biotechnology Inc.) for 2 h at room temperature. For the detection of the protein bands, the ECL Western Blotting Detection System (Amersham Biosciences, Little Chalfont, UK) was used.

### 4.13. Real-Time PCR

SH-SY5Y cells were seeded in a 6-well plate at a density of 2 × 10^5^ cells/well. The cells were incubated with TGF-β for 24 h. Total RNA was isolated using the TRIzol reagent (Invitrogen) and then used for cDNA synthesis, which was performed with the PrimeScript™ RT reagent kit (TaKaRa, Shiga, Japan). The specific genes were quantified using a 7500 Real-Time PCR System (Applied Biosystems, Foster city, CA, USA), with the Power SYBR^®^ Green PCR Master Mix and TaqMan^®^ Gene Expression Master Mix (Applied Biosystems). The sequences of the primers were as follows: human TH, forward 5′-GCTCCACAAGTGTCATCACCTG-3′ and reverse 5′-CCTGTACTGGAAGGCGATCTC-3′; and human GAPDH (Endogenous Control), forward 5′-TGAAGACGGGCGGAGAGAAAC-3′ and reverse 5′- TGACTCCGACCTTCACCTTCC-3′. The PCR was run for 40 cycles at 95 °C (15 s) and 60 °C (1 min). The relative expression levels of the target genes were calculated using the ΔΔCt method, where Ct is the threshold concentration.

### 4.14. Statistical Analysis

Statistical analyses were performed using GraphPad Prism 6 (GraphPad Software, La Jolla, CA, USA). In addition to the forced swimming test (FST); elevated plus maze (EPM); OFT; rotarod test, PAT; von Frey test; comparison of serum LDH and CK were performed as an F test. All data were analyzed using Student’s *t*-test and expressed as mean ± standard deviation. *p*-values < 0.05 were considered significant.

## Figures and Tables

**Figure 1 ijms-22-02580-f001:**
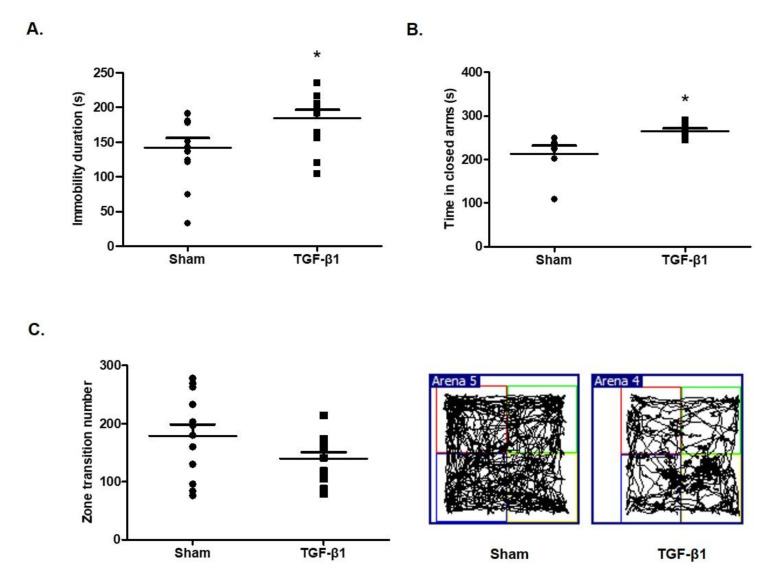
Effects of transforming growth factor (TGF)-β1 on depressive and anxious behaviors in mice. The C57BL/6 mice were intracranially injected with TGF-β1 (100 ng/mice) and the following measurements were performed: (**A**) Immobility duration time by forced swimming test; (**B**) spending time in the closed arms in an elevated plus maze test; (**C**) zone transition numbers in an open field test. Data are presented as mean ± SEM (N = 8–10). * *p* < 0.05 compared with the sham group.

**Figure 2 ijms-22-02580-f002:**
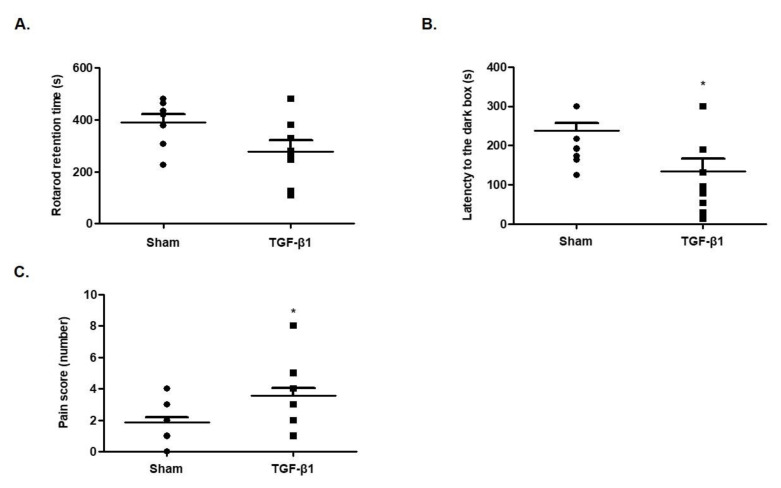
Effects of TGF-β1 on motor function, learning memory and allodynia. (**A**) C57BL/6 mice were intracranially injected with TGF-β1 and a rotarod test was performed to measure retention time on a rotating rod. (**B**) The latency time to enter the dark compartment from the bright compartment using the passive avoidance test. (**C**) The sensitivity to pain was measured by counting the number of evasion behaviors from hair stimulation on hind paw in the von Frey test. Data are presented as mean ± SEM (N = 8–10). * *p* < 0.05 compared with the sham group.

**Figure 3 ijms-22-02580-f003:**
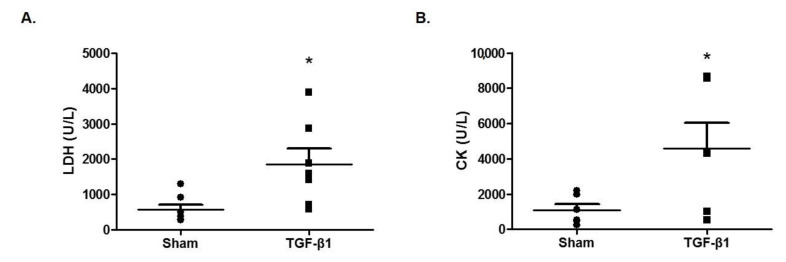
Effects of TGF-β1 on lactate dehydrogenase (LDH) and creatine kinase (CK) in serum. The serum was isolated from C57BL/6 mice at 18 days after TGF-β1 intracerebroventricular (i.c.v.) injection. The serum levels of (**A**) LDH and (**B**) CK were measured using a biochemical analyzer. Data are presented as mean ± SEM (N = 6). * *p* < 0.05 compared with the sham group.

**Figure 4 ijms-22-02580-f004:**
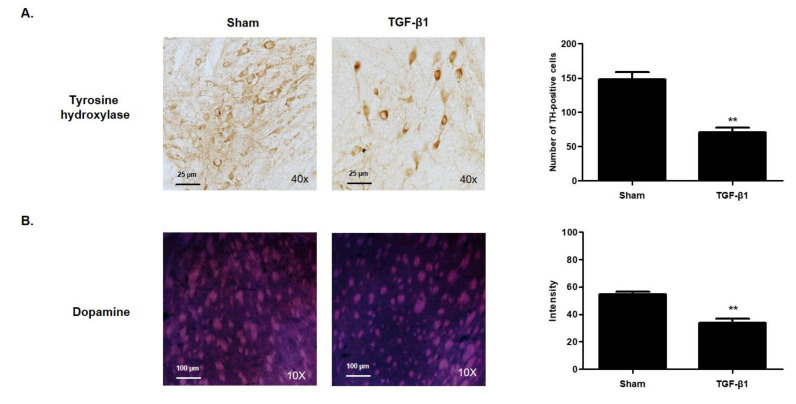
Effects of TGF-β1 on dopaminergic system. The brain was isolated from C57BL/6 mice at 18 days after TGF-β1 i.c.v. injection. (**A**) The level of tyrosine hydroxylase (TH) in the ventral tegmental area was observed using immunohistochemistry staining. (**B**) Dopamine in the striatum was identified using immunofluorescence staining. Representative photomicrographs were taken at magnifications of 40× and 10×. ** *p* < 0.01 compared with the sham group.

**Figure 5 ijms-22-02580-f005:**
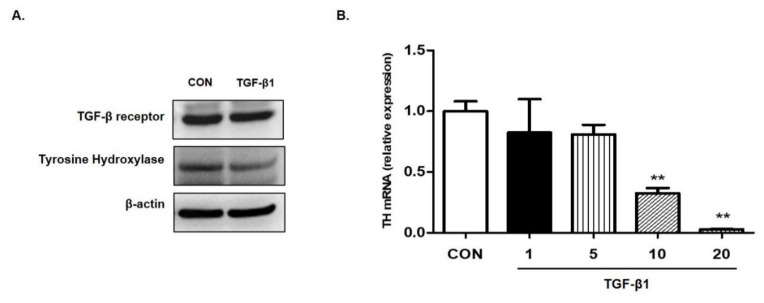
Effect of TGF-β1 on tyrosine hydroxylase in SH-SY5Y neuroblastoma cells. (**A**) The cells were treated with TGF-β at 10 ng/mL concentrations for 72 h. Cell lysates were analyzed using Western blotting with tyrosine hydroxylase antibodies. (**B**) The cells were treated with TGF-β at indicated concentrations for 24 h. Total RNAs were extracted and expression of tyrosine hydroxylase was detected by real-time PCR. GAPDH was used as a loading control. Data are presented as mean ± SD from three independent experiments. ** *p* < 0.01 compared with control.

## Data Availability

The datasets used and/or analyzed in the current study are available from the corresponding author upon reasonable request.

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
