# Peer review of "Exogenous Transforming Growth Factor-β in Brain-Induced Symptoms of Central Fatigue and Suppressed Dopamine Production in Mice"

_ijms, 2021, doi:10.3390/ijms22052580_

Round 1

Reviewer 1 Report

No description of control conditions makes it impossible to review the scientific merits.

Author Response

We appreciate your comment. Your opinion made the quality of our manuscript improved

The mice in control group were received vehicle solution with same volume using same methods as TGF-β group. To make easy to understand, we changed the group name from Control to Sham. And we also described the sham group in the section of 4.2 Intracerebroventricular injection of TGF- β1. The following is the added sentence in the section.

The sham group was treated with an equivalent volume of vehicle (0.1% BSA in PBS) via the same methods used for the TGF-β1 group.

Reviewer 2 Report

This report would be enhanced by improvement in the English.

the experiments and the conclusions are plausible and this report would be a valuable addition to the body of work investigating CFS.

Author Response

We appreciate your kind suggestion.

According to your comment, language of manuscript was checked using the English editing system of this journal.

Round 2

Reviewer 1 Report

The authors addressed my previous comments on the missing description of the control/sham group appropriately. I also noticed the authors improved overall writings.
This study aims to develop an animal model for chronic fatigue syndrome (CFS). The paper now was well written. However, the results do not support the conclusion. Results showed that experimental mice had increased mobility duration and pain sensitivity and reduced tyrosine hydroxylase production. These symptoms are presented in all disorders/diseases with central fatigue, including but not limited to multiple sclerosis, traumatic brain injury, Parkinson's disease, and fibromyalgia. The hallmarks of CFS are post-exertional malaise and unrefreshing sleep. Therefore, the animal is more appropriate to name as an animal model of central fatigue but not CFS or CFS symptoms.

Author Response

Thank you for your kind suggestion. The quality of manuscript is much more improved with your excellent opinion.

According to your opinion, we changed the word, ‘CFS’ to ‘central fatigue’ or ‘fatigue’ in the parts where we concluded our results. In addition, we changed the title. We marked the changes with red color.

Title

Before : Exogenous Transforming Growth Factor-β in Brain-Induced Symptoms of Chronic Fatigue Syndrome and Suppressed Dopamine Production in Mice

After : Exogenous Transforming Growth Factor-β in Brain-Induced Symptoms of Central Fatigue and Suppressed Dopamine Production in Mice

Abstract (the last sentence)

Before : These results, which show TGF-β1 induced CFS-like behaviors by suppressing dopamine production, suggest that TGF-β1 plays a critical role in the development of chronic fatigue syndrome and is, therefore, a potential therapeutic target of the disease.

After : These results, which show TGF-β1 induced fatigue-like behaviors by suppressing dopamine production, suggest that TGF-β1 plays a critical role in the development of central fatigue and is, therefore, a potential therapeutic target of the disease.

Introduction (the last sentence)

Before : In this study, we performed an intracerebroventricular (i.c.v.) injection of TGF-β into the brains of mice and investigated the effect of TGF-β on CFS-like behaviors and the production of neurotransmitters in the brain. Our results suggest that TGF-β1 modulates the behaviors and symptoms of CFS and that this animal model is applicable in a study of the CFS.

After : In this study, we performed an intracerebroventricular (i.c.v.) injection of TGF-β into the brains of mice and investigated the effect of TGF-β on fatigue-like behaviors and the production of neurotransmitters in the brain. Our results suggest that TGF-β1 modulates the behaviors and symptoms related with CFS and that this animal model is applicable in a study of the central fatigue.

Results (the last sentence of section 2.6)

Before : The results indicate that TGF-β1 may induce CFS-like behaviors in mice by regulating the production and secretion of dopamine.

After : The results indicate that TGF-β1 may induce fatigue-like behaviors in mice by regulating the production and secretion of dopamine.

Discussion (the last sentence)

Before : Therefore, this study suggests a potential molecular mechanism that induces CFS with an exogenous injection of TGF-β can be a useful animal model for a pharmaceutical investigation of CFS.

After : However, the effects of TGF-β on characteristic symptoms of CFS, post-exertional malaise and unrefreshing sleep, were remained to be elucidated. Therefore, this study suggests a potential molecular mechanism that induces central fatigue with an exogenous injection of TGF-β can be a useful animal model for a pharmaceutical investigation of CFS.

Round 3

Reviewer 1 Report

The author addressed my concerns appropriately.